# The conformational wave in capsaicin activation of transient receptor potential vanilloid 1 ion channel

Fan Yang[1,2], Xian Xiao[2,3], Bo Hyun Lee[2,4], Simon Vu[2], Wei Yang[1], Vladimir Yarov-Yarovoy[2] & Jie Zheng [2]

The capsaicin receptor TRPV1 has been intensively studied by cryo-electron microscopy and functional tests. However, though the apo and capsaicin-bound structural models are available, the dynamic process of capsaicin activation remains intangible, largely due to the lack of a capsaicin-induced open structural model and the low occupancy of the transition states. Here we report that reducing temperature toward the freezing point substantially increased channel closure events even in the presence of saturating capsaicin. We further used a combination of fluorescent unnatural amino acid (fUAA) incorporation, computational modeling, and rate-equilibrium linear free-energy relationships analysis (Φ-analysis) to derive the fully open capsaicin-bound state model, and reveal how the channel transits from the apo to the open state. We observed that capsaicin initiates a conformational wave that propagates through the S4–S5 linker towards the S6 bundle and finally reaching the selectivity filter. Our study provides a temporal mechanism for capsaicin activation of TRPV1.

[1] Department of Biophysics and Kidney Disease Center, First Affiliated Hospital, Institute of Neuroscience, National Health Commission and Chinese Academy of Medical Sciences Key Laboratory of Medical Neurobiology, Zhejiang University School of Medicine, Hangzhou 310058 Zhejiang Province, China. [2] Department of Physiology and Membrane Biology, University of California, Davis, CA 95616, USA. [3] Institute for Basic Medical Sciences, Westlake Institute for Advanced Study, Westlake University, Shilongshan Road No. 18, Xihu District, Hangzhou 310024 Zhejiang Province, China. [4]Present address: University of Washington, Department of Physiology and Biophysics, Seattle, WA 98195, USA. These authors contributed equally: Fan Yang, Xian Xiao, Bo Hyun Lee. Correspondence and requests for materials should be addressed to F.Y. (email: fanyanga@zju.edu.cn) or to J.Z. (email: jzheng@ucdavis.edu)

Most proteins, including membrane proteins like ion channels[1,2], carry out biological functions through conformational rearrangements. Depending on the thermodynamic stability under varying conditions, not all individual conformational states are always readily accessible by structural biology methods. The transient receptor potential vanilloid 1 (TRPV1) channel, a polymodal receptor for various stimuli such as noxious heat, capsaicin, and proton[3], is an important pain sensor[4] and validated target for anesthetic drugs[5,6]. To understand the activation mechanism of TRPV1 for future drug developments, previously we have employed fluorescent probes to monitor conformational dynamics in one particular region (outer pore domain) of this channel during heat and $Mg^{2+}$ induced activation in living cells[7,8]. Similar imaging-based approaches have been applied to study conformational changes in other channels such as voltage-gated potassium channels[9]. To improve the fluorescent probes, unnatural amino acids such as 3-(6-acetylnaphthalen-2-ylamino)-2-aminopropanoic acid (ANAP) have been developed[10] and employed to study TRPV1 and other ion channels[11–13]. Moreover, with the recent breakthrough in structural biology techniques, high-resolution static structural models of TRPV1 have been determined under cryo conditions[14–16]. All these advancements enable us to investigate the spread of conformational rearrangements within the channel protein when activated by its classic agonist capsaicin in living cells.

In the closed-state cryo-EM structure of TRPV1 (PDB ID: 3J5P), both the S6 bundle crossing and selectivity filter are in a conformation likely too narrow for ions and water molecules to pass[15] (Fig. 1a). In the presence of capsaicin, which adopts a Tail-up and Head-down configuration in the binding pocket and stabilizes the outward conformation of S4–S5 linker[17], the pore radius at the S6 bundle crossing increases to about 2 Å (PDB ID: 3J5R) to allow ion permeation[14] (Fig. 1a). However, whereas this structural model successfully guided the search for capsaicin-binding mechanism[17–22], it exhibits a nearly identical conformation of the selectivity filter as in the closed state, with a pore radius being less than 1 Å. Indeed, in all-atom molecular dynamics and bias-exchange metadynamics simulations, ion permeation can only be observed and studied in the double-knot toxin- and RTX-activated open state (PDB ID: 3J5Q)[23,24], for which both the S6 bundle crossing and selectivity filter are in an open conformation[14]. These studies further suggest that unlike voltage-gated potassium channels where $K^+$ ions are dehydrated to permeate[25], cations passing through TRPV1 are partially hydrated, implicating that the selectivity filter of TRPV1 needs to be physically widened from the observed closed state to allow ion permeation. Moreover, because restrictions to ion permeation at both the selectivity filter and the S6 bundle crossing are serially connected, both of them need to be relaxed to allow for ion flux. Therefore, questions remain regarding how capsaicin binding opens both the selectivity filter and the S6 bundle crossing of

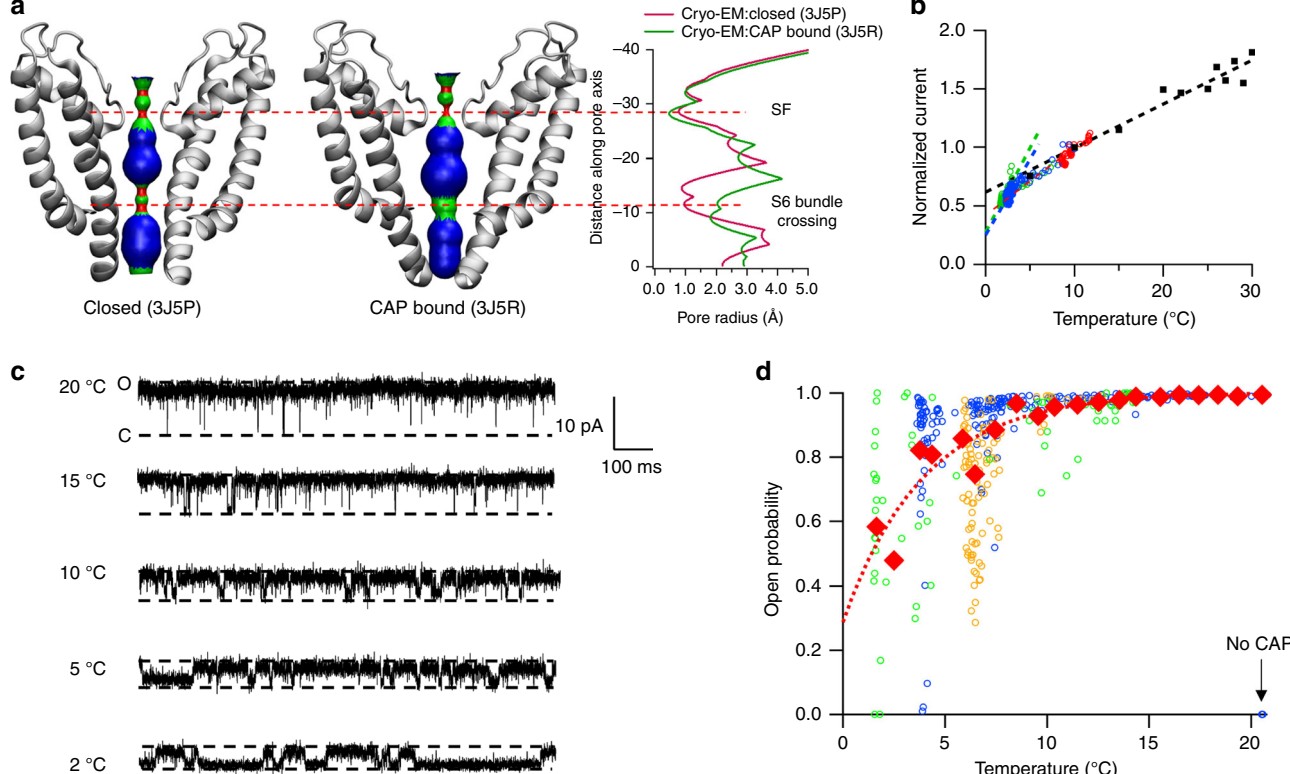

**Fig. 1** TRPV1 closure induced by cooling in the presence of saturating concentrations of capsaicin. **a** Distribution of pore radii in the cryo-EM-derived closed state (3J5P) and capsaicin bound state (3J5R). The regions in red are too narrow to allow a water molecule to pass. The regions in green or blue are sites allowing single or multiple water molecules and ions to pass. Pore radii were calculated by the HOLE program[65]. SF, selectivity filter. **b** Temperature-dependent reduction of macroscopic current amplitude is steeper than that of single-channel conductance. Macroscopic currents from three inside-out patch recordings (shown in red, blue, and green) and single-channel current (shown in black) are normalized to the amplitude at 10 °C. Superimposed are linear fits of the single-channel conductance levels between 5 and 30 °C (black dashed line) or the macroscopic currents in the middle (red dashed line) and low (blue and green dashed lines) temperature ranges. **c** Example single-channel traces at representative temperatures in the presence of 10 μM capsaicin. **d** Representative consecutive open probability measurements of three single-channel recordings (green, blue, and orange circles), as well as the average Po (red diamonds) fitted to an exponential function (red dotted curve)

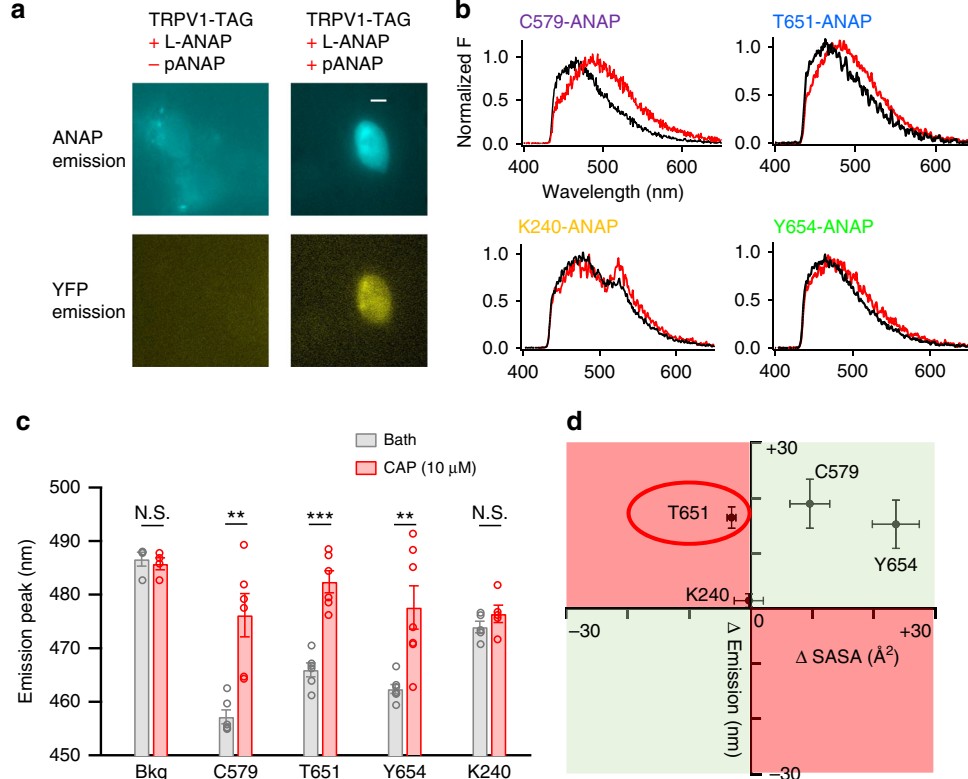

**Fig. 2** Conformational changes near the selectivity filter revealed by ANAP fluorescence. **a** Representative images of negative control cells (for which the pANAP vector was not added) and positive cells expressing ANAP-incorporated TRPV1. YFP (yellow fluorescent protein) fluorescence was readily detected in positive cells but not in negative controls. Negative control cells were imaged with longer exposure time to ensure that no genuine fluorescent signal was missed. Pseudo colors for ANAP and YFP were used. Scale bar: 10 μm. **b** Representative emission spectra (black, bath solution; red, solution containing 10 μM capsaicin) of ANAP incorporated at different sites. For K240 site, the emission peak around 512 nm likely represented YFP emission due to FRET with ANAP[11]. **c** ANAP emission peak values in the absence or presence of capsaicin. Data points were shown as circles. Two-sided $t$-test: **, $p < 0.01$; ***, $p < 0.001$; N.S., not significant; $n = 5$–7. **d** Correlation between shift in ANAP emission peak ($y$ axis) and changes in SASA measured from cryo-EM structures ($x$ axis). Residues in the first and third quadrants (shaded in green) exhibited matched shift in emission and change in SASA, whereas T651 in the second quadrant (shaded in red) showed a discrepancy between these two measurements; $n = 5$–7. All statistical data are given as mean ± s.e.m

TRPV1 to fully open the channel. In the present study, we find that upon binding, capsaicin initiates a conformational wave that propagates through the S4–S5 linker toward the S6 bundle to open this restriction site. The selectivity filter further undergoes conformational rearrangements, which finally lead to ion permeation through the channel.

## Results

**Cooling led to channel closure in the presence of saturating capsaicin.** When TRPV1 structure models were determined under cryo conditions[14,15], even in the presence of full agonist capsaicin, the selectivity filter showed a conformation virtually identical to that in the apo state (Fig. 1a). The observation led us to wonder that, while capsaicin can sustain a fully activated TRPV1 at room temperature, whether it can do so at low temperatures close to the freezing point. We first recorded macroscopic currents in the presence of 10 μM capsaicin using inside-out patch clamp recording while the temperature was reduced from room temperature down to 1.5 °C (Fig. 1b). The slope of temperature-dependent current decline was compared to that of single-channel conductance. We observed that the decline of macroscopic current was steeper, especially at temperatures below 5 °C, suggesting that the channel open probability might have decreased at lower temperatures. To verify this, we conducted single-channel recordings under similar conditions (Fig. 1c). We found that indeed the channel exhibited a dramatic decrease in

the open probability, dropping below 50% even before the temperature reached 0 °C (Fig. 1d). These results revealed that occupancy of ligand-binding sites by capsaicin is insufficient to keep TRPV1 in the fully open state at low temperatures, a conclusion consistent with the similarity in cryo-EM conformations of the selectivity filter in the apo (closed) and capsaicin-bound states.

**ANAP revealed capsaicin-induced conformational changes.** To test whether capsaicin can induce conformational change in the outer pore region, we genetically incorporated a small, environment-sensitive, fluorescent unnatural amino acid (fUAA) 3-(6-acetylnaphthalen-2-ylamino)-2-aminopropanoic acid (ANAP)[10,11,26] into TRPV1 at representative sites (Fig. 2 and Supplementary Figure 1a). Local conformational changes were monitored by the shift in ANAP emission spectrum[10]. We fused a yellow fluorescent protein (YFP) to the C terminus of TRPV1 to facilitate identification of cells expressing ANAP-incorporated full-length channels (Fig. 2a). As it is possible that ANAP incorporation could allosterically alter the ligand gating of TRPV1, we first confirmed using both calcium imaging and patch-clamp recording that all ANAP-incorporated channels included in this study were still activated by capsaicin, though their activation kinetics may have been altered by ANAP incorporation (Supplementary Figure 1b–e). To further minimize any potential ANAP incorporation-induced changes in capsaicin sensitivity, we used a

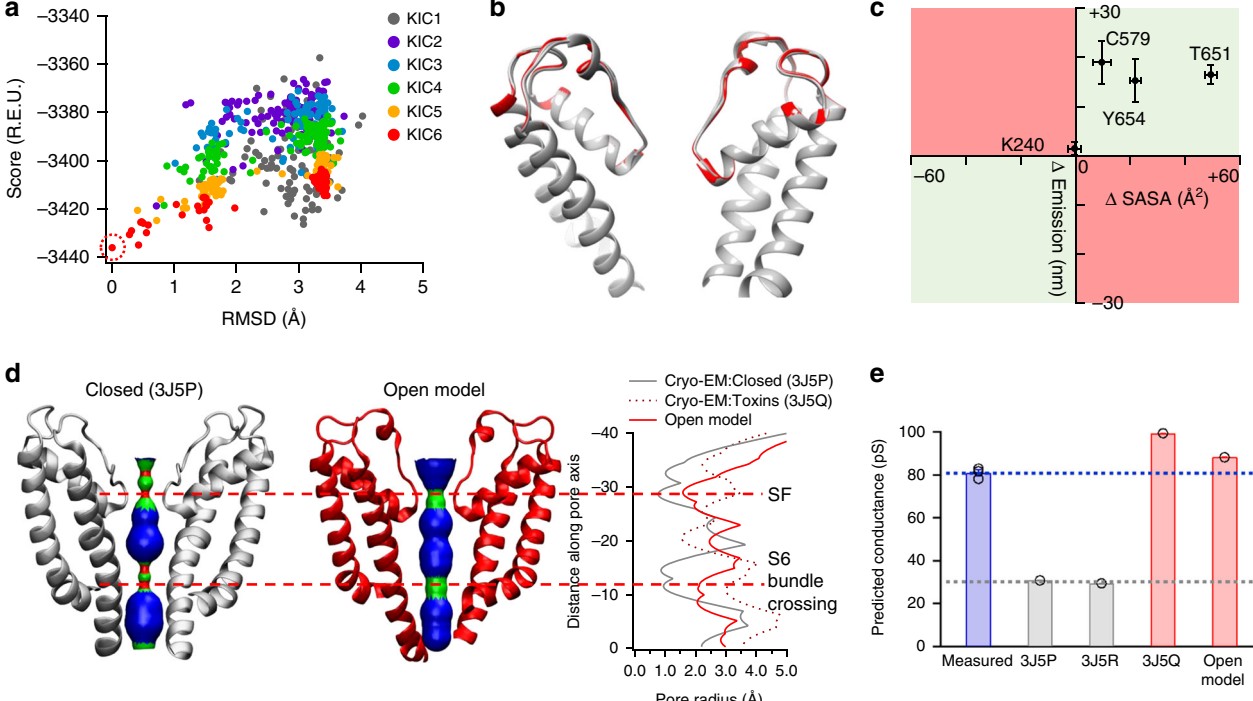

**Fig. 3** Modeling of the capsaicin-induced fully open state. **a** The models after six rounds of KIC loop modeling exhibited a funnel-shaped distribution of total energy calculated by Rosetta (R.E.U., Rosetta energy unit). **b** Top five models after the sixth round of loop modeling were well converged. The model with the lowest energy (red) was chosen as the open-state model. **c** Correlation between shifts in ANAP emission peak (y axis) and changes in SASA measured from our open-state model (x axis); $n = 5–7$. **d** Distribution of pore radii in the cryo-EM-derived closed state (3J5P) and our open-state model. In our model, both the selectivity filer region and the S6 bundle crossing are wide enough to allow permission of water molecules and ions. **e** Conductance of TRPV1 in different states predicted by the HOLE program. Our open-state model was predicted to have similar conductance as values measured by patch-clamp recording with 10 μM capsaicin[17] (the bar in blue). The DkTx and RTX bound state structure (3J5R) determined by cryo-EM represented a fully open state, which shows a predicted conductance close to that of our open-state model and experimentally measured values. The predicted conductance of either the closed state (3J5P) or capsaicin bound state (3J5R) was much smaller (bars in gray). Data points are shown as circles. All statistical data are given as mean ± s.e.m

high capsaicin concentration (10 μM) to produce maximal channel activation without detectable non-specific effects on lipid membrane[27] in the following ANAP experiments. As a positive control, ANAP was attached to C579 on the S4–S5 linker that undergoes an outward movement during capsaicin activation[14,17], hence a capsaicin-dependent change in local environment would be expected. Indeed, the emission peak of ANAP was substantially right-shifted by $19.0 ± 4.4$ nm ($n = 6$) upon capsaicin binding (Fig. 2b–d). As a negative control, ANAP was incorporated at the previously reported K240 site in the intracellular N terminus[11], where there was little shift in the ANAP emission peak during capsaicin activation (Fig. 2b–d). Interestingly, in the outer pore region, ANAP at both T651 and Y654 sites exhibited a larger than 10 nm right-shift in emission peak during capsaicin activation (Fig. 2c, d), indicating that there are clearly detectable conformational changes near the selectivity filter leading to a conductive state.

The right-shift in ANAP emission peak at both C579 and Y654 sites indicated that the local environment might be more hydrophilic upon capsaicin activation[10]. This is in agreement with changes in solvent accessible surface area (SASA) at these sites directly measured from cryo-EM structures in the apo and capsaicin-bound states (Fig. 2d, first quadrant shaded in green). However, though ANAP fluorescence at the T651 site also predicted an increase in exposure to hydrophilic environment, cryo-EM structures showed a slight decrease in SASA (Fig. 2d, second quadrant shaded in red). This discrepancy again indicated that the capsaicin-bound state model from cryo-EM (3J5R) may

not represent the true pore conformation under physiological conditions.

**Open-state model showed conformational changes**. We took an integrative approach to explore the fully open state. Taking the increase in SASA at T651 site as an experimentally derived constraint and 3J5R as a starting template, we used computational tools in the Rosetta suite[28] to search for a stable open pore conformation. After six rounds of kinematic closure (KIC) loop modeling of the selectivity filter region[29], the models exhibited a funnel-shaped energy distribution (Fig. 3a) with high convergence shown in the top four lowest-energy models (Fig. 3b). These models form Cluster 1 among the top ten models with lowest energy values (Supplementary Figure 2a and 2b, color in red), whereas other two clusters of models were observed among the top ten models (Supplementary Figure 2c and 2d; Supplementary Data 1). We chose the model with the lowest energy (Fig. 3b, red; Supplementary Data 2), which shows comparable quality assessment metrics as models derived from cryo-EM studies (Supplementary Table 1), as the final model. This model was further globally refined in Rosetta. The refined model exhibited similar Fourier shell correlation with the cryo-EM density map (capsaicin-bound state, EMD ID: 5777) as the published model (PDB ID: 3J5R) (0.345 and 0.343, respectively; Supplementary Figure 3). When the rotamer library of ANAP was generated (Supplementary Figure 4a and 4b; Supplementary Data 3) and this residue was incorporated at the 651 position

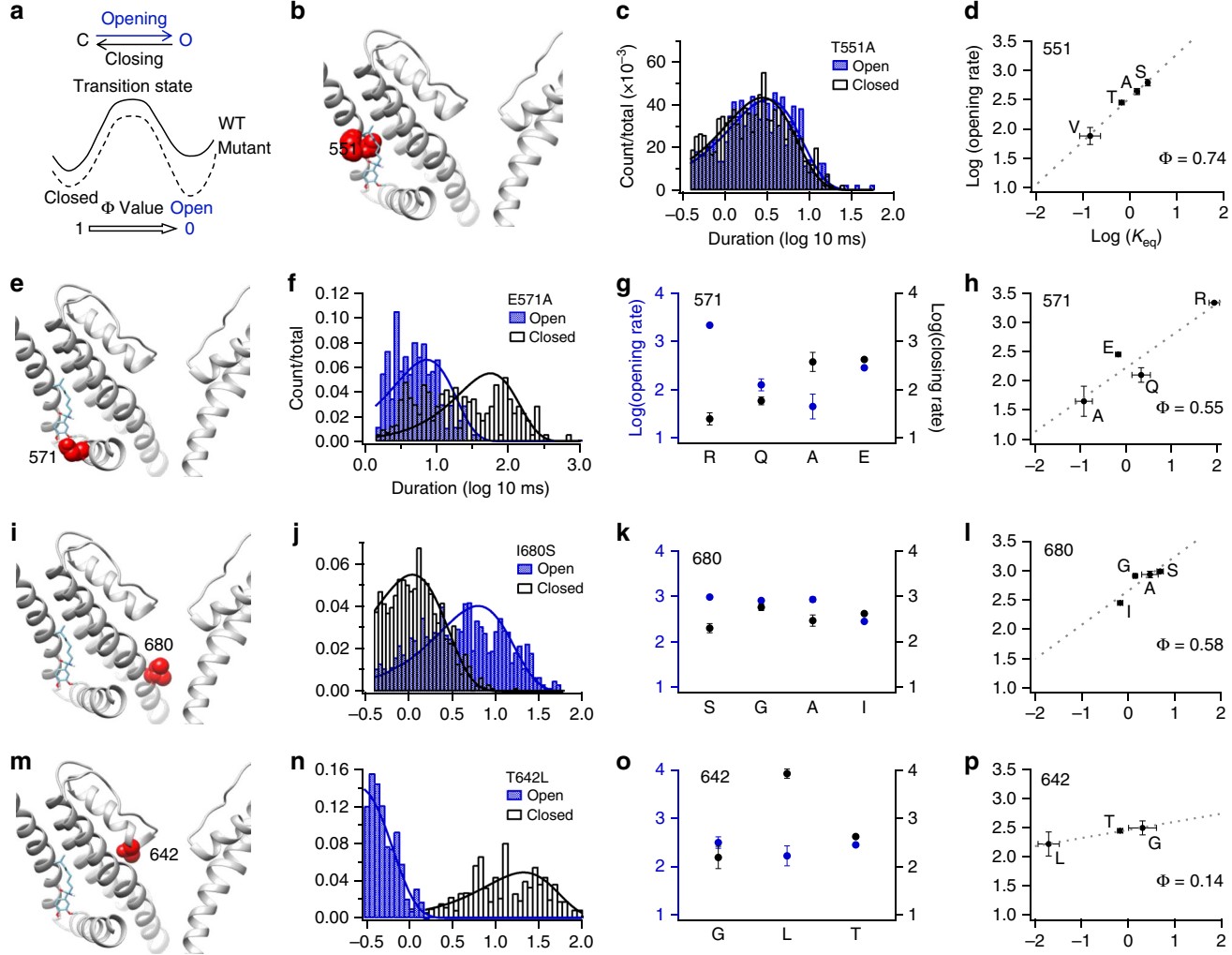

**Fig. 4** Φ-analysis of representative residues on TRPV1. **a** The free energy landscape of a closed-to-open transition. Point mutations, depending on its location, have asymmetrical effects on the free energies of closed and open state, which can be reflected in the kinetics of single-channel events. **b** T551 site (red) locates within the capsaicin-binding pocket. **c** Distribution of the opening (blue) and closing (gray) events duration in the log scale. **d** Brønsted plot to determine the Φ value for the T551 site. For each mutation at the same site, 3–5 independent single-channel recordings were analyzed. **e–h** The E571 site on the S4–S5 linker shows an intermediate Φ value of 0.55. **i–l** The I680 site near the S6 bundle crossing also shows an intermediate Φ value of 0.58. **m–p** The T642 site near the selectivity filter shows a low Φ value of 0.14. All statistical data are given as mean ± s.e.m

(Supplementary Figure 4c and 4d), its sidechain exhibited a significant increase in SASA in our open model as compared to that in the cryo-EM-derived closed-state model (Supplementary Figure 4e and 4f; Supplementary Data 4 and 5), confirming that our capsaicin-induced open-state model is compatible with the experimentally observed right shift in ANAP emission peak (Fig. 2c, d).

In this model, both T651 and Y654 sites near the selectivity filter show a positive change in SASA that correlates with the right-shift in ANAP emission peak (Fig. 3c, first quadrant). Furthermore, the pore radii at both the selectivity filter and the S6 bundle crossing were sufficiently large (~2 Å) to permit ion permeation (Fig. 3d). The anticipated conductance of our open model and models derived from cryo-EM studies was predicted by a structure-based Ohmic model in the HOLE program, which had successfully predicted conductance of ion channels such as the acetylcholine receptor and Maltoporin[30]. Indeed, the open model showed a predicted conductance similar to the experimentally determined value during capsaicin activation[17] (Fig. 3e), supporting that our model represents a fully open and conductive state. In contrast, models in Cluster 2 and Cluster

3 showed both smaller pore radii and smaller predicted conductance as compared to the open model (Supplementary Figure 2e and 2f).

**Φ-analysis revealed a comformational wave.** Both modeling and ANAP measurement indicated a widespread of conformational changes induced by capsaicin binding. To reveal the conformational wave during this dynamic process, we performed Φ-analysis[31,32] that reports the asymmetrical energetic effects on the closed and open states by point-mutations at various channel structures (Fig. 4a). For example, mutations to residues in the ligand-binding pocket are expected to exhibit a stronger impact on the initial gating transition, whereas mutations to residues at the activation gate are expected to exhibit a stronger impact on the final gating transition; the asymmetrical energetic effect can be quantified when the transition rates between the closed state and the open state were measured from single-channel recordings of the mutant channels. A Φ value falls between zero and one, whose physical interpretation is obviously model dependent[33]. For instance, if there were several parallel activation

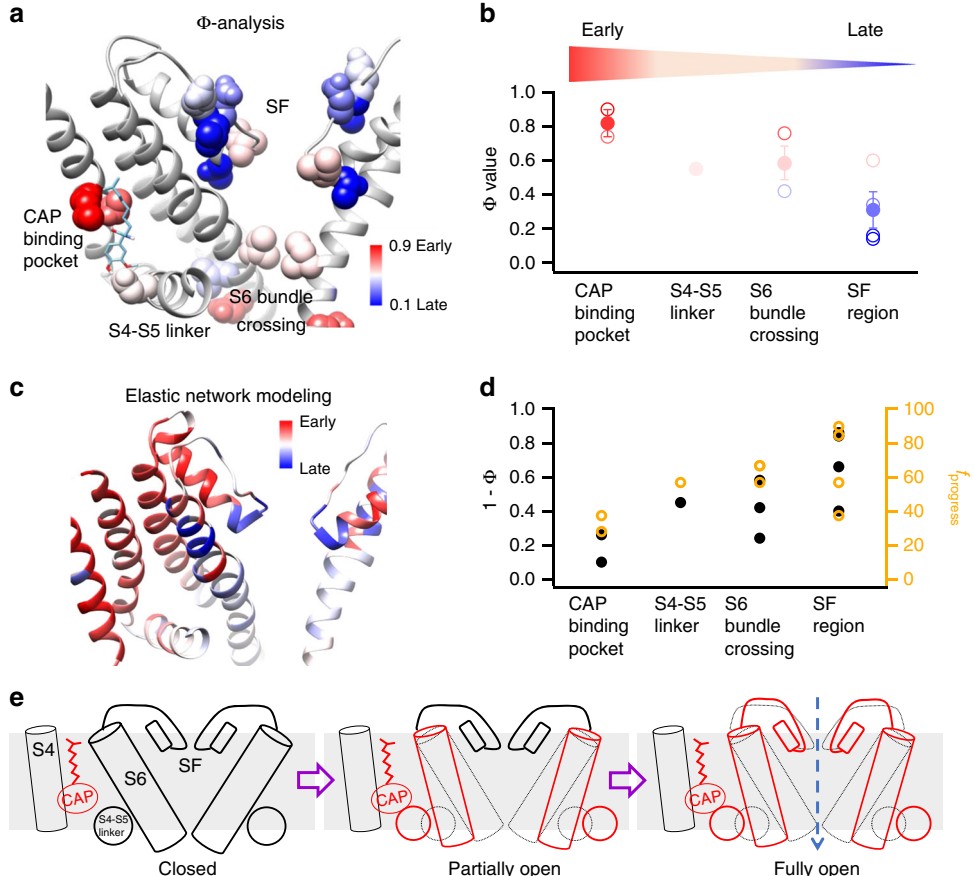

**Fig. 5** The conformational wave in capsaicin activation of TRPV1. **a** Measured Φ values were mapped on TRPV1 structure. The sites moved early (large Φ values) or late (small Φ values) were colored in red or blue, respectively. **b** The Φ values were clustered and averaged by their locations. Capsaicin-binding pocket (M548 and T551): 0.82 ± 0.08; S4–S5 linker (E571): 0.55; S6 bundle crossing (I680, L682, and E685): 0.59 ± 0.10; Selectivity filter (T642, I643, D647, L648, and E649): 0.31 ± 0.11. **c** $f_{progress}$ values calculated by iENM were mapped on TRPV1. Residues moved early (small $f_{progress}$ value) or late (large $f_{progress}$ value) were colored in red or blue, respectively. **d** The Φ values (black) and $f_{progress}$ values (orange) matched well with each other in multiple domains. **e** A schematic diagram showing the temporal sequence of events upon capsaicin binding that lead to the full activation of TRPV1. All statistical data are given as mean ± s.e.m

pathways between the closed and open states, the Φ value for a particular position in any pathway would be affected by the probability that activation takes the pathway. As a simplified approach, though, the Φ value could be taken to reflect the relative position of the transition state along a single reaction pathway: if a residue moves earlier during capsaicin activation, then its Φ value would be closer to unity; a residue moving later has a Φ value closer to zero. Our previous study has suggested that the capsaicin molecule first forms a hydrogen bond with T551 on the S4, followed by another hydrogen bond with E571 on the S4–S5 linker[17]. We found that the Φ value of T551 (0.74) inside the capsaicin-binding pocket was larger than that of E571 (0.55) on the S4–S5 linker (Fig. 4b–h), confirming that indeed the T551 site moved earlier than E571. Moreover, we observed that for the M548 site, which is also inside the binding pocket, the Φ value (0.90) was very close to unity (Supplementary Figure 5). Large Φ values at M548 and T551 clearly indicate that capsaicin initiates a conformational wave starting from its binding pocket. Previous Φ-analysis studies have shown that, for the acetylcholine receptor[32] and the cystic fibrosis transmembrane conductance regulator (CFTR)[34], conformational waves are also initiated from ligand-binding pockets.

To reveal how the capsaicin-induced conformational wave propagates, we performed Φ-analysis on more residues throughout the channel. We observed that residues around the S6 bundle crossing (Fig. 4 and Supplementary Figure 5), such as I680 (Fig. 4i–l, Φ = 0.58), exhibited Φ values (0.59 ± 0.10, n = 3 (I680, L682 and E685); Fig. 5b, in pink) similar to that of E571 on the S4–S5 linker. This observation suggests that the S4–S5 linker and the S6 bundle crossing move simultaneously as a unit, which is not surprising given the strong coupling between these domains both structurally and functionally in TRPV1 as well as voltage-gated ion channels[35,36]. In contrast, Φ values of residues near the selectivity filter were much smaller (0.34 ± 0.09, n = 5 (T642, I643, D647, L648, and E649); Fig. 5b, in blue). For instance, T642 has a Φ value of 0.14 (Fig. 4m–p). This indicates that the selectivity filter region indeed participates in capsaicin-induced activation, and it moves later.

When all measured Φ values were mapped onto the structure of TRPV1 (Fig. 5a), we identified a clear pattern: residues in the capsaicin-binding pocket show high Φ values close to one, residues in the S4–S5 linker and the S6 bundle crossing exhibited intermediate and similar Φ values, whereas Φ values were small for residues near selectivity filter (Fig. 5b). Such a clustered and graded pattern of Φ value distribution suggests that upon capsaicin binding, a conformational wave is initiated from the ligand-binding pocket, which propagates toward the S6 bundle crossing through the S4–S5 linker and finally reaches the selectivity filter region.

To corroborate the Φ-analysis, we employed interpolated elastic network modeling (iENM)[37] to find the transition path

from the closed state (3J5P) to the fully open state (Fig. 3d). In this coarse-grained modeling method, residues of the channel protein were represented by spheres interconnected with springs. The transition path was calculated by solving the saddle points of a general potential function composed of the two harmonic potentials of the starting and ending states. This method predicted temporal movements in the acetylcholine receptor[38] and HCN channel[39] that agreed with structural and functional tests. In our case, the distribution of $f_{progress}$ values calculated from iENM, which reflects the temporal sequence of residue movements, matched well with the experimentally measured Φ values (Fig. 5c, d).

## Discussion

Our results from multiple approaches revealed that capsaicin activation of TRPV1 is not limited to opening of the S6 bundle crossing; instead, the whole pore including the selectivity filter undergoes conformational rearrangements. These findings are consistent with the standing view that TRPV1 pore exhibits high flexibilities[40]. Cryo-EM studies showed that the S6 bundle crossing and selectivity filter constitute two restrictions for ion permeation[14–16]. The S6 bundle crossing was previously seen to undergo state-dependent conformational changes to control ion accessibility of residues inside the pore[41], similar to that observed in Shaker channels[42,43]. The selectivity filter of the closely related TRPV2 channel also exhibits conformational plasticity[44]. Existing studies have yielded evidence supporting the possibility of a selectivity filter gate in many ion channels, including prokaryotic potassium channels[45], cyclic nucleotide-gated channel[46], $Ca^{2+}$-activated potassium (BK) channel[47], and shaker potassium channel[48,49]. Indeed, when we decreased the temperature toward the freezing point, even a saturating concentration of capsaicin did not sustain the open state (Fig. 1b–d). Given that the capsaicin molecule was still observed in its binding pocket and the S6 bundle crossing was in an open conformation under cryo conditions[14] (PDB ID: 3J5R), our observation most likely suggests that cooling may lead to closure of the gate at selectivity filter, which is compatible with the cryo-EM structures where this region exhibited nearly identical conformations in both apo and capsaicin-bound states. Moreover, the selectivity filter and S6 need to be functionally coupled when they both form a restriction for permeation; in the known cases, substantial S6 movements appear to serve the role of transmitting activation stimuli to induce apparently subtle movements in the selectivity filter. Our observed distribution of Φ values is in full agreement with the scenario where after capsaicin molecules are bound to TRPV1, a conformational wave spreads first horizontally to S6, then vertically to the selectivity filter (Fig. 5e).

The selectivity of TPRV1 has been shown to exhibit use-dependent changes during capsaicin activation to allow permeation of large cations[50], which enables the channel as a molecular portal for delivery of anesthetic drugs like QX-314[51]. The distinct conformations of this region observed in both cryo-EM studies[14–16] and our experiments may serve as the structural basis for this phenomenon and guide drug development using TRPV1 as a delivery portal in future.

The recent revolution in cryo-EM techniques[52], as demonstrated in the TRPV1 channel[14–16], has led to an explosion in the number of high-resolution structures of proteins in defined states. How a protein dynamically transits from one state to another is critical to understand its function yet difficult to be examined, because the low occupancy (high energy) of the transition states makes these states rarely captured and resolved by structural biology methods. Moreover, such a transition often occurs at or below microsecond time scale as revealed by single-channel

recordings[53,54], which is beyond the time resolution of most imaging-based techniques. Observations of capsaicin binding[17–19] and capsaicin-induced conformational wave in TRPV1 demonstrate the power of combining computational modeling with functional tests (such as thermodynamic mutant cycle analysis, fUAA, and Φ-analysis) to reveal dynamic processes in proteins. Such an integrative approach is generally applicable to ion channels, which constitute the third largest family of small-molecule drug targets among all proteins[55].

## Methods

**Molecular biology and cell transfection.** Murine TRPV1 (a gift from Dr. Michael X. Zhu, University of Texas Health Science Center at Houston) was used in this study. eYFP was fused to the C terminus of TRPV1 to help identify channel-expressing cells. Tagging of eYFP did not change the functional properties of TRPV1 as we have reported before[56]. Point mutations were generated by QuickChange II mutagenesis kit (Agilent Technologies). Primers used in this study were summarized in Supplementary Table 2. All mutants were confirmed by DNA sequencing.

HEK293T cells were purchased from American Type Culture Collection (ATCC). These cells were authenticated to be contamination-free by ATCC. Cells were cultured in a Dulbecco's modified eagle medium with 10% fetal bovine serum and 20 mM L-glutamine at 37 °C with 5% $CO_2$. cDNA constructs of channels were transiently transfected with Lipofectamine 2000 (Life technologies) according to the manufacturer's protocol. One or 2 days after transfection, electrophysiological recordings were performed.

**Chemicals.** All chemicals were purchased from Sigma-Aldrich unless otherwise stated.

**Fluorescence unnatural amino acid.** L-ANAP methyl ester was purchased from AsisChem. pANAP vector was purchased from Addgene. ANAP was incorporated into TRPV1 with a TAG amber stop codon mutation as previously reported[10]. Briefly, after co-transfection of both TRPV1 and pANAP vectors, ANAP was directly added to the culture medium to the final concentration of 20 μM. After 1–2 days, ANAP-containing culture medium was completely changed. Cells were further cultured in ANAP-free medium overnight before experiments.

ANAP fluorescence was excited by the wLS LED light source (QImaging) with a 375/28 excitation filter, T400lp dichroic mirror, and 435LP emission filter on an inverted fluorescence microscope (Nikon TE2000-U) using a 40× oil-immersion objective (NA 1.3). Emission spectrum of ANAP was imaged with an Acton SpectraPro 2150i spectrograph in conjunction with an Evolve 512 EMCCD camera. The emission peak value was found by fitting the spectrum with a skewed Gauss distribution.

**Electrophysiology.** Patch-clamp recordings were performed with a HEKA EPC10 amplifier controlled by PatchMaster software (HEKA). Whole-cell recordings at ±80 mV were used to test whether an ANAP-incorporated channel was functional. Patch pipettes were prepared from borosilicate glass and fire-polished to resistance of ~4 MΩ. For whole-cell recording, serial resistance was compensated by 60%. For single-channel recordings in Φ-analysis, patch pipettes were fire-polished to a higher resistance of 6–10 MΩ. To maximize the chance of obtaining a patch with only one channel, single-channel recordings were performed about 8 h after transfection. A solution with 130 mM NaCl, 10 mM glucose, 0.2 mM EDTA, and 3 mM Hepes (pH 7.2) was used in both bath and pipette for either whole-cell or single-channel recordings. Membrane potential was clamped at +80 mV for single-channel recordings. Current was sampled at 10 kHz and filtered at 2.9 kHz. All recordings were performed at room temperature (22 °C) with the maximum variation of 1 °C.

Capsaicin was perfused to membrane patch by a gravity-driven system (RSC-200, Bio-Logic). Bath and capsaicin solution were delivered through separate tubes to minimize the mixing of solutions. Patch pipette was placed in front of the perfusion tube outlet.

**Molecular modeling.** To model the capsaicin-induced fully open state of TRPV1, membrane-symmetry-loop modeling was performed using the Rosetta molecular modeling suite[28] version 2015.25. Starting with the cryo-EM structure of capsaicin bound state (3J5P), the selectivity filter, the pre-S6 linker, and the S1–S2 linker were modeled de novo with the KIC loop modeling protocol[57,58]. About 10,000–20,000 models were generated each round. These models were first filtered by the SASA value at T651 site. Only the models with an increase in SASA larger than 20 Å² compared to the closed state were allowed to pass. Among the filtered models, the top 20 models by energy were selected as the inputs for next round of loop modeling. After six rounds of KIC loop modeling, the top five models converged well. The model with the lowest energy was finally selected as the open-state model. This model was further refined by the relax application[59] within the Rosetta suite. The FSC between the model and the capsaicin bound state electron density map (EMD ID: 5777) was also calculated by Rosetta.

To model the sidechain conformation of ANAP within the TRPV1 models, we generated the rotamer library of ANAP as described[60]. Briefly, the chemical structure of ANAP was optimized by Gaussian version 09[61]. Then the backbone-dependent rotamer library was generated by the MakeRotLib application[60] in Rosetta. Then the ANAP residue was incorporated into TRPV1 models in the apo and open states by the Backrub application[62,63] in Rosetta. For each state, 5000 models were generated and the model with the lowest energy was selected. The rotamer library of ANAP is attached as a Supplementary Data 1.

Command lines used in Rosetta to perform the modeling processes are attached in Supplementary Methods. SASA of each residue in TRPV1 structures models was measured by RosettaScripts[64] within the Rosetta suite. The scripts to perform SASA measurements and filtering are also attached in Supplementary Methods.

Pore radius of a TRPV1 model was calculated by the HOLE program[65] version 2.0[66].

All the molecular graphics of TRPV1 models were rendered by UCSF Chimera[67] software version 1.12[68].

**Elastic network modeling**. iENM was performed using iENM web server[37] (http://enm.lobos.nih.gov). Transmembrane domains of same TRPV1 subunit in closed-state structure (PDB ID: 3J5P) and our capsaicin-induced open-state model were submitted as the starting and ending conformation, respectively. The distance cutoff for elastic interaction between alpha carbon atoms was set as 13 Å. Based on this cutoff, two harmonic potentials were constructed for the starting and ending conformations, respectively. The server solved the saddle points of a general potential function composed of these two harmonic potentials. The calculated $f_{progress}$ values reflected the temporal sequence of movements.

**Φ-analysis**. We followed the principle of Φ-analysis described in published literatures in detail[31,32]. All Φ-analysis was performed on the I574A background channel, where maximum Po is reduced to allow accurate measurements[17]. To ensure saturation of binding in our experiments, 30–100 μM capsaicin was used. At such a high concentration, only the patches showing the activities of one channel was used for analysis. Single-channel data was processed by the QuB software[69] version 1.4[70]. Opening and closing events were detected during idealization with the half amplitude method. Using the simple close ↔ open model (extended data Fig. 2A), the forward rate and backward rate were determined after imposing a deadtime of 0.3–0.4 ms. The equilibrium constant was calculated as the ratio of forward and backward rates. Opening rates and equilibrium constants for each mutant at the same residue site were plotted on log scale (Brønsted plot) before fitting with a linear function to determine the Φ value.

**Statistics**. All experiments have been independently repeated for at least three times. All statistical data are given as mean ± s.e.m. Two-sided Student's t-test was applied to examine the statistical significance. N.S. indicates no significance; ** and ***, $p < 0.05$ and $p < 0.001$, respectively.

## Data availability

Data supporting the findings of this manuscript are available from the corresponding authors upon reasonable request.

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

## Acknowledgements

We are grateful to Sharona Gordon for the K240 construct, Andy Watkins for technical assistance with ANAP rotamer library generation, and to our lab members for assistance and insightful discussions. This work was supported by funding from NIH (R01NS072377 to J.Z., R01NS103954 and R56NS097906 to J.Z. and V.Y.-Y.), AHA (14POST19820027 to F.Y., 16PRE29340002 to S.V., and 16PRE26960016 to B.H.L.), the National Basic Research Program of China (2013CB910204 to W.Y.), the Natural Science Foundation of China (81571127 to W.Y. and 31741067 to F.Y.), and Young Thousand Talent Program of China to F.Y. This work was also supported by the bioinformatics computation platform in Zhejiang University School of Medicine.

## Author contributions

F.Y., X.X., B.H.L., and S.V. conducted the experiments including patch-clamp recording, mutagenesis, imaging, molecular modeling, and data analysis; V.Y.-Y. supervised molecular modeling and revised the manuscript; J.Z., and F.Y. prepared the manuscript; J.Z., V.Y.-Y., W.Y., and F.Y. conceived and supervised the project, participated in data analysis and manuscript writing.

## Additional information

**Competing interests:** The authors declare no competing interests.

