## [Peer Review File · Nature Communications]

Reviewers' comments:

Reviewer #1 (Remarks to the Author):

The manuscript by Xiao et al. reports an interesting investigation of the gating mechanism of TRPV1. The paper focuses on the so-called "conformational wave", i.e. a structural description of the progressive transition from the closed to the open state. The goal of the investigation is to shed light on a fundamental issue that is still very poorly understood, namely the allosteric coupling between capsaicin binding and gate opening. Although I find these results novel and potentially interesting for a wide audience of scientists, I have concerns about the interpretation of some of the results discussed and these prevent me from recommending publication of this manuscript. In particular, I have the following objections:

1) The authors claim that, in the experimentally derived structural model of TRPV1 in complex with capsaicin, the pore radius at the level of the selectivity filter is too small to allow ion conduction. I find the argument weak, the pore radius is known to be a poor descriptor of conductance, especially in the selectivity filter. For instance, in potassium selective channels, the selectivity filter is so narrow to accommodate only a single file of waters or ions and yet the channel is conductive. This type of claims can be based only on all atom calculations: by computing of the free energy profile for permeation of a cation, one could objectively make predictions about the expected kinetics for ion crossing and thus make an inference on the functional state corresponding to the structure. In absence of these calculations, it is difficult to judge whether the expectation on the conductance expressed by the authors is reasonable or not.

2) The authors find another piece of information in support of the non-conductive nature of the capsaicin-bound structure: T651 is predicted, on the basis of the ANAP spectrum, to be more solvent-exposed in the open than in the closed state. Since the capsaicin-bound experimental structure suggests the opposite, the authors conclude that the structure must be wrong. However, the ANAP spectrum responds, of course, to all sorts of environmental changes. For instance, T651 can show differential interactions with protein amino acids with formation and breaking of H-bonds, can interact with the cations resident in the selectivity filter, with lipids at the membrane interface, etc. A reliable interpretation of the peak shift is impossible without explicit modeling of the unnatural amino acid in the context of the protein structure.

3) I appreciate the fact that the de novo modeling of the loop results in better quality metrics compared to the experimentally derived model. However, the modeling approach used here is not global, i.e. it assumes that the rest of the structure is good and keeps it fixed. In other words, if the old model contained bad contacts between the loop and, let us say S6, the refinement would remove them by remodeling the loop. How can we be sure that it was not S6, instead, to be in the wrong conformation? In addition to that, the authors never discuss how the new model fits within the electron density found experimentally? Does the refinement improve the fit? A global refinement and a comparison with the electron density are both needed to judge objectively the issue.

4) I have reservations about the interpretation of the phi-analysis presented in this paper, i.e. as a spatio-temporal map of the conformational transition. My concerns are mostly related to the phi values around 0.5 for which, I argue, an unambiguous interpretation is impossible. Let us start from the interpretation proposed for the small phi values measured in the case of T642. Here mutation to L produces a large decrease of the equilibrium constant and of the closing rate, while leaving the opening rate unaffected (the mutation to G is hardly statistically significant). In a two-state model these facts can be confidently associated to a selective destabilization of the open state, closed and transition states are untouched so the residue should be crucial only at the very end of the pathway. A similar line of reasoning shows that in the case of T551 both the open and the transition states are affected, therefore this residue should be crucial to stabilize the transition state and thus involved very early in the pathway. However, the interpretation of the values close

to 0.5 is problematic: a simultaneous stabilization of the closed state and destabilization of the open one has the same effect as a destabilization of both the transition and open states. While in the latter case, one would interpret the $\phi=0.5$ as indicative of the fact that the transition state is halfway in the pathway (i.e. it is, in part, similar to the open state), in the former scenario such interpretation would be certainly less appropriate. It is interesting to note that one of the $\phi=0.5$ cases is that of E571 for which charge inversion (E571R) increases significantly the equilibrium constant, increases the opening rate and decreases the closing rate. A possible interpretation is that E571 is engaged in a favorable electrostatic interaction in the closed state and unfavorable in the open one. When the charge is inverted (E571R), the opposite is true: unfavorable interaction in the closed and favorable in the open, i.e. nothing happens to the transition state and the residue is crucial at both the beginning and at the end of the transition pathway. Overall, I think that the interpretation of these ϕ values is really not solid enough. Each claim should be extensively backed up by other experimental approaches, like all atom free energy calculations, cysteine cross-linking, etc.

Reviewer #2 (Remarks to the Author):

Thank you for the opportunity to provide feedback on 'The conformational wave in capsaicin activation of TRPV1 ion channel' by Xiao and colleagues. This paper can be summed up in two parts. First, the paper makes an effort to predict and validate a fully open conformation of TRPV1. Second, the paper uses ϕ analysis to interrogate the temporal sequence of conformational changes that occur between the capsaicin binding site and channel opening (in the S6 and eventually selectivity filter regions). In the first part of the paper, the prediction of the open state conformation (particularly with respect to the adoption of a selectivity filter conformation) is based on experimental constraints arising from shifts in the fluorescence emission peak of ANAP, arising from environmental changes in different channel conformations. The second part of the paper, related to the conformational wave of channel activation, involves single channel recording from a series of mutations at various position, and deriving a ϕ value that describes to what extent a change in open state stability can be attributed to changes in the opening rate. The general findings of the ϕ value study describe a sequence of conformational changes beginning in the capsaicin binding site, propagating to the S4/S5 linker and S6, and finally into the selectivity filter. I found that the paper was interesting to read, but I did not find that the different questions addressed were particularly complimentary to one another. Please consider the following comments that I feel would improve the paper.

Major comments:

1. The paper was quite short on details and discussion, and I think the journal format would have allowed for deeper description of methods and data.
2. I found the description of the computation modeling to be quite opaque. There was no description of the interpolated elastic network modeling approach. Also, by my reading, the primary constraint available and used by the authors was the solvent accessibility of residue Thr651 (based on ANAP spectra). An important caveat here is that the solvent accessibility reported by ANAP may be significantly different from the native Thr, as they are quite different in structure. Based on inspection of the cryo-EM structures, accessibility of amino acids in this region of the channel seem that they could be quite sensitive to sidechain size and conformation. Also on this point, the raw data showed altered responses to capsaicin and 2-APB in several of these mutants (Extended data fig. 1C,D,E). The authors did not comment on these differences – would they suggest that regions involved in gating in the ANAP mutants are not functioning as in the WT channel?
3. The ϕ value analysis was interesting although some positions were not especially sensitive to mutation, and this leads to greater uncertainty in terms of the slope calculation. I would have appreciated a more in depth discussion of the possible meaning of the variable ϕ values calculated for selectivity filter positions (eg. 643 vs. 648). It was also not clear to me how this

finding, although challenging to collect, would influence general thinking about the operation of these channels.

Minor comments:

1. Line 138 Refs 34, 35: Although I have always found the data in these papers quite interesting, I think it is a stretch to say that the idea of a selectivity filter activation gate in Shaker and closely-related channels has been embraced by the field.
2. Line 143: typos on : 'Horizontally and virtically'

Reviewer #3 (Remarks to the Author):

At the request of the handling editor, I examined specifically the computational/modelling aspects of this work, and I will limit my comments to them.

Overall, this work is a nice example of a judicious combination of experimental and computational methods, and the choice of methods seems perfectly adequate in view of the authors' objectives. My suggestions for improvements concern mainly the presentation of the methods and results.

Integrative modelling: The limiting factor in random search procedures such as the one employed in this work is the amount of sampling. If only the smallest-energy structure is selected for analysis, as done here, the question arises if it is really a low-energy structure or merely the best one found during a perhaps insufficient sampling procedure. One way to generate more trust in the result is to look at more than one low-energy structure. If several structures lead to similar conclusions, these conclusions are much more credible than conclusions drawn from a single structure that might well be an accident of bad sampling. It would also be nice if the authors could make available a few low-energy structures as supplementary material to allow readers to do their own inspection. In any case, I consider it compulsory to publish the "best" structure in machine-readable form (e.g a PDB or mmCIF file), since it is one of the main results of this study.

iENM analysis: Given the limitations of both ENMs in principle and their specific use for identifying transition pathways, I would tone down a bit the assertion that iENM analysis provides an independent test of the results of Φ -analysis, since the word "test" suggests that iENM yields more reliable answers than Φ -analysis. ENMs work best for large proteins with rigid domains and flexible connecting regions. They don't work very well for proteins with an overall homogeneous density, such as transmembrane helix bundles. The protein studied here lies somewhere in between these two extremes, so the reliability of any ENM approach is hard to assess. Moreover, iENM, like any transition path search based only on the initial and final structures, cannot be expected to provide reliable information on anything but the very first and the very last parts of a multi-step transition process, nor to yield reliable results in the eventuality of multiple distinct transition paths. It is still of interest to compare the results of iENM and Φ -analysis, but I wouldn't call one method a test for the other one.

Computational reproducibility: The description of the computations that were performed is much too incomplete to allow a reader to judge if the techniques were applied with sufficient care or to allow verification by reproduction. To reach today's quality standards for reproducible research, the following information must be supplied by the authors:

1) **Software citation:** all software used must be cited with (1) a reference to the source code and the precise version that was used and (2) a citation of the paper that describes the software. This applies to Rosetta (paper cited but no reference to the precise version of the code), HOLE (paper cited but no reference at all to the code and version) and QuB (no reference at all). Only Chimera is cited satisfactorily.

2) All software (scripts, workflows, ...) written specifically for this study must be published and referenced (zenodo.org or figshare.com are good options), or at least provided as supplementary material with the article. This applies to the Rosetta scripts used for molecular modelling, and to the scripts used for the statistical analyses.

3) The iENM analysis is not mentioned at all in "Materials and Methods". If any software was used or written for this, the two preceding paragraphs apply. If the Web server at nih.gov was used, please:

a) Say so clearly, and cite the URL.

b) Provide the distance cutoff parameter that was used.

c) Provide the two input structures that were used (PDB code is OK for unmodified files from the PDB, otherwise supply the files)

4) SASA measurements on the cryo-EM structures are discussed in several places, but no explanation is given for how they were obtained. Neither the method applied nor the software used are mentioned.

Reviewer 1

1) The authors claim that, in the experimentally derived structural model of TRPV1 in complex with capsaicin, the pore radius at the level of the selectivity filter is too small to allow ion conduction. I find the argument weak, the pore radius is known to be a poor descriptor of conductance, especially in the selectivity filter. For instance, in potassium selective channels, the selectivity filter is so narrow to accommodate only a single file of waters or ions and yet the channel is conductive. This type of claims can be based only on all atom calculations: by computing of the free energy profile for permeation of a cation, one could objectively make predictions about the expected kinetics for ion crossing and thus make an inference on the functional state corresponding to the structure. In absence of these calculations, it is difficult to judge whether the expectation on the conductance expressed by the authors is reasonable or not.

We agree with the reviewer that generally speaking pore radius does not accurately predict conductance and have revised the manuscript to soften our stand. Of the three available cryo-EM structures of TRPV1, the capsaicin-bound structure (PDB ID: 3J5R) is almost identical to the apo (therefore closed) channel (PDB ID: 3J5P) in the selectivity filter region but differs in the S6 conformation, whereas the DkTx/RTX-bound structure (PDB ID: 3J5Q) showed a different selectivity filter conformation^{1,2}. All-atom molecular dynamics and bias-exchange meta-dynamics simulations have been employed to study ion permeation of TRPV1 (now cited in our manuscript)^{3,4}, however, statistical analysis of permeation events could only be performed on the DkTx and RTX bound state model (3J5Q) but not on the capsaicin bound state (3J5R) likely due to the small radius of pore near the selectivity filter. Indeed, the pore radius at the selectivity filter in the DkTx and RTX bound state model (> 2.0 Å) is much larger than that in the capsaicin bound state (< 1.0 Å, which is too narrow to conduct even fully dehydrated ions (Fig. 1A and Fig. 2D). For instance, the potassium ions (1.33 Å in radius⁵), which are permeant to TRPV1 channel as demonstrated in patch-clamp recordings⁶, will be too large to go through the selectivity filter in the capsaicin bound state model (3J5R). In comparison, the smallest pore radius observed in the selectivity filter of potassium channels such as KcsA channel (1K4C) and Kv1.2/2.1 chimera channel (2R9R) is 1.60 Å and 1.65 Å, respectively⁷. These observations strongly suggest that the capsaicin-bound state model is in a non-conducting state. Furthermore, the all-atom molecular dynamics and bias-exchange metadynamics simulations showed that, in contrast to the permeation mechanism of Kv channels where the potassium ions have to be dehydrated to go through the selectivity filter⁵, cations permeate through TRPV1 with partial hydration^{3,4}. Therefore, the selectivity filter of an open TRPV1 channel is expected to be large enough to accommodate the cations and associated water molecules.

In addition, we have investigated this issue with an experimental approach: we directly measured the open probability of TRPV1 channels in the presence of saturating concentrations of capsaicin at low temperatures. We observed at both macroscopic and single-channel levels that, as the recording temperature approached zero degree, the open probability started to drop sharply. The open probability fell below 50%, that is, the majority of the channels would be in closed states even before reaching the cryo condition used for

structural studies. These new data are now presented in the new Figure 1 of the revised manuscript.

We used the structure-based Ohmic model within the HOLE program to predict the conductance of TRPV1 channel; this method successfully predicted the conductance of ion channels with a relatively large pore like the acetylcholine receptor and Maltoporin⁸. In the present study, the predicted conductance of TRPV1 was compared with experimentally measured values: we found that the predictions and measurements for the open state were in good agreement, and the apo state and capsaicin bound state models showed much smaller conductance (Fig. 3e and Supplementary Figure 2f).

In summary, based on cryo-EM studies, all-atom molecular dynamics and bias-exchange metadynamics simulations, functional recordings at low temperatures, and our structure-based Ohmic model calculation, the most likely scenario is that the capsaicin-bound state model (3J5R) is in a non-conducting conformation. At the very least, the cryo-EM studies and our ANAP experiments clearly showed that the selectivity filter of TRPV1 can adopt different conformations, and our Φ analysis showed that such conformational change in the selectivity filter occurs later than that in the S6 bundle crossing.

2) The authors find another piece of information in support of the non-conductive nature of the capsaicin-bound structure: T651 is predicted, on the basis of the ANAP spectrum, to be more solvent-exposed in the open than in the closed state. Since the capsaicin-bound experimental structure suggests the opposite, the authors conclude that the structure must be wrong. However, the ANAP spectrum responds, of course, to all sorts of environmental changes. For instance, T651 can show differential interactions with protein amino acids with formation and breaking of H-bonds, can interact with the cations resident in the selectivity filter, with lipids at the membrane interface, etc. A reliable interpretation of the peak shift is impossible without explicit modeling of the unnatural amino acid in the context of the protein structure.

As suggested by the reviewer, we have now explicitly modeled the ANAP conformation within the closed state cryo-EM model and our open state model based on ANAP fluorescence experiments (new Supplementary Figure 4 of the revised manuscript). We first generated a rotamer library of the ANAP sidechain and then integrated ANAP into the models. The conformations of the sidechain and backbone nearby were optimized by the Backrub application in the Rosetta suite. We observed that the SASA of ANAP at the 651 site was indeed increased from the closed state to the open state (new Supplementary Figure 4), which is consistent with the right shift of ANAP emission spectrum we observed. Therefore, the emission peak shift as we observed can be interpreted as a result of conformation changes induced by capsaicin activation of TRPV1 channel.

3) I appreciate the fact that the de novo modeling of the loop results in better quality metrics compared to the experimentally derived model. However, the modeling approach used here is not global, i.e. it assumes that the rest of the structure is good

and keeps it fixed. In other words, if the old model contained bad contacts between the loop and, let us say S6, the refinement would remove them by remodeling the loop. How can we be sure that it was not S6, instead, to be in the wrong conformation? In addition to that, the authors never discuss how the new model fits within the electron density found experimentally? Does the refinement improve the fit? A global refinement and a comparison with the electron density are both needed to judge objectively the issue.

We apologize that the modeling procedure in the submitted manuscript was not clearly described (a point also raised by Reviewer #3), and have addressed this issue during the revision. The modeling sections in both main text and methodologies are substantially revised. To clarify, we did perform global refinements with our model (new Supplementary Figure 3 of the revised manuscript) by the relax application in Rosetta, after kinematic closure (KIC) loop modeling. The refined model was fitted to the capsaicin bound state electron density map (EMD ID: 5777; new Supplementary Figure 3a and 3b). Despite that the selectivity filter region of our model is in a different conformation from the capsaicin bound state model (PDB ID: 3J5R), when compared to the capsaicin bound state electron density map (EMD ID: 5777), our relaxed model exhibits a FCS value similar to that of model 3J5R as calculated by Rosetta (0.345 and 0.343, respectively). Therefore, we believe that the quality of our model is not worse than the experimentally derived ones.

4) I have reservations about the interpretation of the phi-analysis presented in this paper, i.e. as a spatio-temporal map of the conformational transition. My concerns are mostly related to the phi values around 0.5 for which, I argue, an unambiguous interpretation is impossible.

We agree that interpretation of the phi analysis is indeed model dependent, and hence in the present study we followed the established approach of multiple published studies and analyzed experimental observations based on a simple $C \leftarrow \rightarrow O$ transition, so as to avoid interpreting details that couldn't be constrained by experimental data. While phi values around 0.5 are more likely affected by models, as the reviewer pointed out, our major goal was to establish the beginning and end of the capsaicin-induced gating process, and our novel finding in the present study was that the end process occurs at the selectivity filter. Regardless of the way of data interpretation, as agreed by the reviewer, our data clearly indicate that the residues in the capsaicin binding pocket (such as T551) move at the beginning and the residues at and near the selectivity filter (such as T642) move at last. In response to the reviewer's comments, we have added more discussions on the interpretation of the Φ analysis in the revised manuscript, including the caveats in interpreting data derived from this method.

The Φ analysis is a powerful experimental approach in that it probes the transition process in proteins, such as the opening and closing of an ion channel, that occur within the time scale of a few micro-seconds or faster⁹. The information obtained by the Φ analysis is not easily accessible to other techniques that reflect steady state properties of a protein like cysteine cross-linking for the following reasons: (1) Cross-linking events report close

proximity (within 15 angstroms or even larger) without providing temporal information. Indeed, it is well recognized that cross-linking may occur during rare events of close proximity because cross-linking, once occurred, is irreversible. (2) Most of the capsaicin-induced conformational changes occur to the transmembrane core domain. No large rearrangement of the scale mentioned above that can be reliably detected by cross-linking is expected in this region. (3) The reviewer may have suggested using cross-linking to force two sites to move together during activation gating. However, it is not clear to us how this would help reveal conformational changes or test our model. In addition, this approach may produce artificial coupling and other non-specific effects that can cover up genuine gating transitions.

We agree with the reviewer that all-atom molecular dynamics simulation can indeed be a powerful approach, however the micro-seconds time scale is still too demanding for regular MD simulation hardware most scientists (including us) can access. All-atom MD simulation of ion channel gating in microseconds time scale was only reported by David Shaw's group using their advanced computation power. Capsaicin-induced TRPV1 activation occurs over about 100 ms, hence all-atom MD was beyond our reach. Partly for this reason, to corroborate the Φ analysis, we employed the coarse-grained interpolated elastic network modeling technique to model the same gating process, which reported a similar temporal order of transitions seen in our Φ analysis (Fig. 5d). The other two complementary approaches used in this study, site-specific ANAP measurements and Rosetta structural modeling, also yielded a consistent picture.

We agree with the reviewer that residues with a Φ value around 0.5 may be ambiguous to interpret. Specifically, for the residue E571, our previous study¹⁰ has shown that when the protein backbone was fixed in the closed state to mimic the initial contact between the capsaicin and protein, a hydrogen bond was first formed between T551 (not E571) and the neck of capsaicin. Upon relaxation of protein backbone, a second hydrogen bond was formed between E571 and the head of capsaicin. Therefore, E571 contacts with capsaicin later than T551. This observation, cited in the manuscript, is in good agreement with the measured order of Φ values for these two residues in the present study.

Reviewer 2

1. The paper was quite short on details and discussion, and I think the journal format would have allowed for deeper description of methods and data.

We have now added more descriptions on the methods and data, in both the main text and supplementary information sections. The revised manuscript also includes new data from both functional recordings and modeling. We appreciate the reviewer's suggestion.

2. I found the description of the computation modeling to be quite opaque. There was no description of the interpolated elastic network modeling approach. Also, by my reading, the primary constraint available and used by the authors was the solvent accessibility of residue Thr651 (based on ANAP spectra). An important caveat here is

that the solvent accessibility reported by ANAP may be significantly different from the native Thr, as they are quite different in structure. Based on inspection of the cryo-EM structures, accessibility of amino acids in this region of the channel seem that they could be quite sensitive to sidechain size and conformation.

We agree with the reviewer. We have now explicitly modeled the ANAP conformation within the closed state cryo-EM model and our open state model based on ANAP fluorescence experiments (Supplementary Figure 4). We first generated the rotamer library of ANAP sidechain and then integrated ANAP with the models. The conformation of the sidechain and backbone nearby was optimized by the Backrub application in the Rosetta suite. We observed that the SASA of ANAP at the 651 site was indeed increased from the closed state to the open state (Supplementary Figure 4e and 4f), which is consistent with the right shift of ANAP emission spectrum we observed. Therefore, the emission peak shift as we observed can be interpreted as a result of conformation changes induced by capsaicin activation of TRPV1 channel.

Also on this point, the raw data showed altered responses to capsaicin and 2-APB in several of these mutants (Extended data fig. 1C,D,E). The authors did not comment on these differences – would they suggest that regions involved in gating in the ANAP mutants are not functioning as in the WT channel?

2-APB is a TRPV1 agonist that activates the channel in a capsaicin-independent manner¹¹, and hence has been used as a way to control for capsaicin activation in our previous investigation of capsaicin-binding pocket¹⁰. The binding site for 2-APB remains unclear. While all the ANAP-incorporated mutant channels in this study exhibited robust capsaicin responses, the 2-APB responses of the two nearby pore mutants (T651 and Y654) were clearly reduced. The reason for this particular change is unknown. To ensure that experimental observations weren't affected by potential changes in ligand sensitivity due to mutations, we used a high capsaicin concentration (10 μ M; about two order of magnitude higher than the EC50 value for wild-type channels) without the non-specific effects on lipid membrane and other channels¹² in the ANAP experiments.

3. The phi value analysis was interesting although some positions were not especially sensitive to mutation, and this leads to greater uncertainty in terms of the slope calculation. I would have appreciated a more in depth discussion of the possible meaning of the variable phi values calculated for selectivity filter positions (eg. 643 vs. 648). It was also not clear to me how this finding, although challenging to collect, would influence general thinking about the operation of these channels.

We have revised the manuscript according to your suggestions. To recap, for Φ analysis it is more important to focus on the general trend of the Φ value distribution, which likely reflects a conformational wave. For instance, when Φ analysis was applied to the acetylcholine receptor to map the conformational wave induced by ligand binding, a graded distribution of Φ values, like what we observed in this study, was observed where most

residues with a high Φ value are clustered in the extracellular ligand binding domain, while most residues with a low Φ value are clustered in the transmembrane domains^{13,14}. However, there were still high Φ value residues found in the M2-M3 linker, which is about 3 nm away from the extracellular ligand binding domain. The low Φ value residues in the transmembrane domains are also quite scattered¹⁴. This type of observations, likely also seen in our present study, may be due to out-of-sync local structural adjustments or even non-specific mutational effects; they highlight the need to look at the whole picture for Φ analysis. The temporal sequence of TRPV1 activation gating driven by capsaicin has not been previously investigated with functional tests. Our present study, by identifying the selectivity filter being a part of the gating machinery, contributes to a more complete picture of the ligand gating process and provides a mechanistic interpretation of the existing structural information.

Minor comments:

1. Line 138 Refs 34, 35: Although I have always found the data in these papers quite interesting, I think it is a stretch to say that the idea of a selectivity filter activation gate in Shaker and closely-related channels has been embraced by the field.

We now present these previous studies as being suggestive of the possibility of a selectivity filter gate in Shaker channels.

2. Line 143: typos on : 'Horizontally and virtically'

We appreciate it and have corrected the text accordingly.

Reviewer 3

One way to generate... It would also be nice if the authors could make available a few low-energy structures as supplementary material to allow readers to do their own inspection. In any case, I consider it compulsory to publish the "best" structure in machine-readable form (e.g a PDB or mmCIF file), since it is one of the main results of this study.

Agreed. We have shown all top 10 models with the lowest energy after the sixth round of kinematic loop modeling in the new Supplementary Figure 2. We observed three clusters of models within the top 10 models. These clusters of models exhibited similar selectivity filter conformation (Supplementary Figure 2a, dashed box in black). Cluster 1 contains top four models with the lowest energy. Cluster 2 and Cluster 3 have two and four models, respectively, that exhibited energy levels slightly higher than that of Cluster 1 (Supplementary Figure 2b, 2c and 2d). Models in Cluster 1 yielded a predicted conductance similar to experimentally measured values (Supplementary Figure 2e and 2f). For these reasons, we chose the model in Cluster 1 with the lowest energy as the final model. The Cluster 1 models were presented in the original Fig. 2B (new Fig. 3b), where

the model in red is the final model. This model has been provided in the supplementary materials. Our model in its PDB format is provided as a supplementary file to the revised manuscript.

iENM analysis: Given the limitation... It is still of interest to compare the results of iENM and Φ -analysis, but I wouldn't call one method a test for the other one.

We agree with the reviewer that iENM is not a test of Φ analysis and have modified the text accordingly.

Computational reproducibility:

1) Software citation: all software used must be cited with (1) a reference to the source code and the precise version that was used and (2) a citation of the paper that describes the software. This applies to Rosetta (paper cited but no reference to the precise version of the code), HOLE (paper cited but no reference at all to the code and version) and QuB (no reference at all). Only Chimera is cited satisfactorily.

We have added information about software in the supplementary information. Thanks for your suggestion.

2) All software (scripts, workflows, ...) written specifically for this study must be published and referenced (zenodo.org or figshare.com are good options), or at least provided as supplementary material with the article. This applies to the Rosetta scripts used for molecular modelling, and to the scripts used for the statistical analyses.

We have now provided all the Rosetta scripts and command lines as a supplementary file.

3) The iENM analysis is not mentioned at all in "Materials and Methods". If any software was used or written for this, the two preceding paragraphs apply. If the Web server at nih.gov was used, please:

- a) Say so clearly, and cite the URL.
- b) Provide the distance cutoff parameter that was used.
- c) Provide the two input structures that were used (PDB code is OK for unmodified files from the PDB, otherwise supply the files)

We have now provided all the information as suggested in the supplementary information.

4) SASA measurements on the cryo-EM structures are discussed in several places, but no explanation is given for how they were obtained. Neither the method applied nor the software used are mentioned.

We have provided all the information, including the Rosetta scripts to perform SASA measurements and filtering, in the supplementary information.

References

- 1 Cao, E., Liao, M., Cheng, Y. & Julius, D. TRPV1 structures in distinct conformations reveal activation mechanisms. *Nature* **504**, 113-118, doi:10.1038/nature12823 (2013).
- 2 Liao, M., Cao, E., Julius, D. & Cheng, Y. Structure of the TRPV1 ion channel determined by electron cryo-microscopy. *Nature* **504**, 107-112, doi:10.1038/nature12822 (2013).
- 3 Jorgensen, C., Furini, S. & Domene, C. Energetics of Ion Permeation in an Open-Activated TRPV1 Channel. *Biophysical journal* **111**, 1214-1222, doi:10.1016/j.bpj.2016.08.009 (2016).
- 4 Darre, L., Furini, S. & Domene, C. Permeation and dynamics of an open-activated TRPV1 channel. *Journal of molecular biology* **427**, 537-549, doi:10.1016/j.jmb.2014.11.016 (2015).
- 5 Doyle, D. A. *et al.* The structure of the potassium channel: molecular basis of K⁺ conduction and selectivity. *Science* **280**, 69-77 (1998).
- 6 Caterina, M. J. *et al.* The capsaicin receptor: a heat-activated ion channel in the pain pathway. *Nature* **389**, 816-824, doi:10.1038/39807 (1997).
- 7 Naranjo, D., Moldenhauer, H., Pincuntureo, M. & Diaz-Franulic, I. Pore size matters for potassium channel conductance. *The Journal of general physiology* **148**, 277-291, doi:10.1085/jgp.201611625 (2016).
- 8 Smart, O. S., Breed, J., Smith, G. R. & Sansom, M. S. A novel method for structure-based prediction of ion channel conductance properties. *Biophysical journal* **72**, 1109-1126, doi:10.1016/S0006-3495(97)78760-5 (1997).
- 9 Zhou, Y., Pearson, J. E. & Auerbach, A. Phi-value analysis of a linear, sequential reaction mechanism: theory and application to ion channel gating. *Biophysical journal* **89**, 3680-3685, doi:10.1529/biophysj.105.067215 (2005).
- 10 Yang, F. *et al.* Structural mechanism underlying capsaicin binding and activation of the TRPV1 ion channel. *Nature chemical biology* **11**, 518-524, doi:10.1038/nchembio.1835 (2015).
- 11 Hu, H. Z. *et al.* 2-aminoethoxydiphenyl borate is a common activator of TRPV1, TRPV2, and TRPV3. *The Journal of biological chemistry* **279**, 35741-35748, doi:10.1074/jbc.M404164200 (2004).
- 12 Ingolfsson, H. I. *et al.* Phytochemicals perturb membranes and promiscuously alter protein function. *ACS chemical biology* **9**, 1788-1798, doi:10.1021/cb500086e (2014).
- 13 Purohit, P., Mitra, A. & Auerbach, A. A stepwise mechanism for acetylcholine receptor channel gating. *Nature* **446**, 930-933, doi:10.1038/nature05721 (2007).
- 14 Purohit, P., Gupta, S., Jadey, S. & Auerbach, A. Functional anatomy of an allosteric protein. *Nature communications* **4**, 2984, doi:10.1038/ncomms3984 (2013).

REVIEWERS' COMMENTS:

Reviewer #1 (Remarks to the Author):

The authors have appropriately addressed all my concerns. I think that the presentation of the results is greatly improved. I recommend publication of the manuscript

Reviewer #2 (Remarks to the Author):

This review is for a resubmitted version of "The conformational wave in capsaicin activation of TRPV1 ion channel".

The authors have addressed my comments satisfactorily in general. The addition of the temperature dependence of open probability was a helpful rationalization of their approach. I felt that a couple of issues should be clarified in the text. In relationship to previous comment #2, the text describes 'normal capsaicin activation' (line 100) of the ANAP-incorporated mutants. I think that this is not quite accurate, as the mutants exhibit extraordinarily slow capsaicin-mediated activation (Supplemental Figure 1b), and one of the mutants generates very small currents (perhaps due to poor expression due to inefficient ANAP incorporation, but this was not explicitly tested) that are also very slow to open and close. Overall, a main concern of this experiment is that the channels are able to open, but a more accurate description of the changes in these mutants would be appropriate.

Another minor point for clarification relates to the statement (lines 239-240):

"... in TRPV1 channels capsaicin binding initiates a conformational wave that spreads first horizontally to S6, then vertically to the selectivity filter (Fig. 5e)". This may reflect a difference in our interpretation of the data, but I would say this statement has a different meaning from saying that a conformational wave is initiated in the capsaicin binding pocket. That is, in the saturating concentrations of the single channel experiments used, the opening and closing events are likely not reflecting capsaicin binding/unbinding as implied by this sentence. Some minor rewording here would help.

Reviewer #3 (Remarks to the Author):

The authors have replied satisfactorily to my comments and provided the missing software references as well as the requested data files.

From my point of view (reminder: I evaluated only the modelling procedures) the article can be published in its current state. However, authors and editors should ensure that the descriptions of the five supplementary datasets will be easily available to readers of the published article. They are not provided in the material for review, I had to request them from the editorial office.

Reviewer 2:

“The authors have addressed my comments satisfactorily in general. The addition of the temperature dependence of open probability was a helpful rationalization of their approach. I felt that a couple of issues should be clarified in the text. In relationship to previous comment #2, the text describes ‘normal capsaicin activation’ (line 100) of the ANAP-incorporated mutants. I think that this is not quite accurate, as the mutants exhibit extraordinarily slow capsaicin-mediated activation (Supplemental Figure 1b), and one of the mutants generates very small currents (perhaps due to poor expression due to inefficient ANAP incorporation, but this was not explicitly tested) that are also very slow to open and close. Overall, a main concern of this experiment is that the channels are able to open, but a more accurate description of the changes in these mutants would be appropriate.”

We agree with the reviewer that when ANAP was incorporated, the TRPV1 mutants exhibited altered capsaicin activation kinetics. As the reviewer pointed out, the key observation here is that these ANAP-incorporated channels remained potently activated by capsaicin, hence allowing the monitoring of capsaicin-induced conformational change with ANAP fluorescence. We have added a more accurate description of these mutants in the text as suggested by the reviewer. We now state that “...we first confirmed that all ANAP-incorporated channels included in this study were still activated by capsaicin using both calcium imaging and patch-clamp recording, though their activation kinetics may have been altered by ANAP incorporation”.

"... in TRPV1 channels capsaicin binding initiates a conformational wave that spreads first horizontally to S6, then vertically to the selectivity filter (Fig. 5e)". This may reflect a difference in our interpretation of the data, but I would say this statement has a different meaning from saying that a conformational wave is initiated in the capsaicin binding pocket. That is, in the saturating concentrations of the single channel experiments used, the opening and closing events are likely not reflecting capsaicin binding/unbinding as implied by this sentence. Some minor rewording here would help.

Indeed, in our Φ analysis the single-channel opening and closing events do not reflect capsaicin binding/unbinding. We have modified this sentence accordingly to avoid any confusion. It is now stated “...the scenario where after capsaicin molecules are bound to TRPV1, a conformational wave spreads first horizontally to S6, then vertically to the selectivity filter”.

Reviewer 3

“From my point of view (reminder: I evaluated only the modelling procedures) the article can be published in its current state. However, authors and editors should ensure that the descriptions of the five supplementary datasets will be easily available to readers of the published article. They are not provided in the material for review, I had to request them from the editorial office.”

We agree with the reviewer that the supplementary datasets should be available to download together with the main text. We will work with the editor to ensure the public availability of these datasets.